# Regression of nonalcoholic fatty liver disease is associated with reduced risk of incident diabetes: A longitudinal cohort study

Dong Hyun Sinn[1,2☯], Danbee Kang[2,3☯], Eliseo Guallar[3,4], Sung Chul Choi[5], Yun Soo Hong[4], Yewan Park[6], Juhee Cho[2,3,4]*, Geum-Youn Gwak[1]*

1 Department of Medicine, Samsung Medical Center, Sungkyunkwan University School of Medicine, Seoul, South Korea, 2 Department of Clinical Research Design and Evaluation, SAIHST, Sungkyunkwan University, Seoul, South Korea, 3 Center for Clinical Epidemiology, Samsung Medical Center, Sungkyunkwan University, Seoul, South Korea, 4 Departments of Epidemiology and Medicine and Welch Center for Prevention, Epidemiology and Clinical Research, Johns Hopkins Medical Institutions, Baltimore, MD, United States of America, 5 Center for Health Promotion, Samsung Medical Center, Sungkyunkwan University, Seoul, South Korea, 6 Department of Internal Medicine, Kyung Hee University Hospital, Seoul, South Korea

☯ These authors contributed equally to this work.
* gy.gwak@samsung.com (G-YG); jcho@skku.edu (JC)

**Data Availability Statement:** The data contain potentially sensitive information that can be used to identify individuals. Also, the data policy at the authors' institution regulates sharing a de-identified

## Abstract

### Objective

Non-alcoholic fatty liver disease (NAFLD) is potentially reversible. However, whether improvement of NAFLD leads to clinical benefits remains uncertain. We investigated the association between regression of NAFLD and the risk of incident diabetes in a longitudinal way.

### Methods

A cohort of 11,260 adults who had NAFLD at ~~in~~ an initial exam, had the second evaluation for NAFLD status at 1~2 years from an initial exam were followed up for incident diabetes from 2001 and 2016. NAFLD was diagnosed with abdominal ultrasound.

### Results

At baseline, NAFLD was regressed in 2,559 participants (22.7%). During 51,388 person-years of follow-up (median 4 years), 1,768 participants developed diabetes. The fully adjusted hazard ratio (HR) for incident diabetes in participants with regressed NAFLD compared to those with persistent NAFLD was 0.81 [95% confidence interval (CI) 0.72–0.92]. When assessed by NAFLD severity, among participants with a low NAFLD fibrosis score (NFS) (< -1.455), participants with regressed NAFLD had a lower risk of incident diabetes than those with persistent NAFLD (HR 0.77, 95% CI 0.68–0.88). However, in participants with an intermediate to high NFS ($\geq$ -1.455), the risk of incident diabetes was not different between NAFLD regression and persistence groups (HR 1.12, 95% CI 0.82–1.51).

data set. Data are available upon request to the Samsung Medical Center Institutional Data Access / Ethics Committee (dm.cha@samsung.com) after approval. Researchers who meet the criteria for access to confidential data may contact the Samsung Medical Center Institutional Data/Ethics Committee or email the corresponding author.

**Funding:** The author(s) received no specific funding for this work.

**Competing interests:** The authors have declared that no competing interests exist.

**Abbreviations:** BMI, body mass index; CIs, confidence intervals; HDL-C, high-density lipoprotein cholesterol; HOMA-IR, Homeostasis Model Assessment of Insulin Resistance; HRs, hazard ratios; LDL-C, low-density lipoprotein cholesterol; NAFLD, nonalcoholic fatty liver disease; TG, triglycerides; US, ultrasound.

## Conclusions

Regression of NAFLD was associated with decreased risk of incident diabetes compared to persistent NAFLD. However, the benefit was evident only for NAFLD patients with low NFS. This suggests that early intervention for NAFLD, before advanced fibrosis is present, may maximize the metabolic benefit from NAFLD regression.

## Introduction

Nonalcoholic fatty liver disease (NAFLD) is a condition in which the liver accumulates fat without significant alcohol intake, viral hepatitis, medications that would cause fatty liver, or other obvious causes [1]. NAFLD is the most common chronic liver disease, with a worldwide prevalence of 25% [2]. NAFLD patients are at an increased risk of adverse outcomes, including overall mortality, and liver-specific morbidity and mortality [3], and are projected to continue to increase, which has an important impact on public health [4]. The clinical burden of NAFLD is not only confined to liver-related outcomes [5]. Hepatic fat accumulation is accompanied by abnormal hepatic energy metabolism [6], and impaired insulin-mediated suppression of hepatic glucose and very low-density lipoprotein production [7], leading to hyperglycemia, hypertriglyceridemia and hyperinsulinemia. Individuals with NAFLD are at a 2.2-fold higher risk of incident diabetes [8].

NAFLD has been associated with persistence or worsening of metabolic abnormalities [9,10]. NAFLD is potentially reversible condition [11]. Reversal of hepatic fat accumulation may lead to metabolic benefit. However, to date, whether regression of NAFLD is associated with a decreased risk of incident diabetes is largely unexplored. Therefore, in this study, we investigated the association between regression of NAFLD and the risk of incident diabetes in a longitudinal way using a large cohort.

## Methods

### Study population

We conducted a retrospective cohort study of men and women of 20 years of age or older who underwent at least 2 health check examinations with abdominal ultrasound (US) between 2001 and 2016 at the Samsung Medical Center (SMC). Since the objective of the study was to evaluate NAFLD regression on incident diabetes, we included subjects with fatty liver on abdominal US at the first exam (N = 25,962). To evaluate change of NAFLD status, we used the results of the second abdominal US exam which took within 1 to 2 years interval after the first abdominal US exam. The time of the second abdominal US exam was considered as baseline visit. Then, the third and any subsequent exams were used to determine the status of diabetes. To control immortal time bias [12], participants who had diabetes in the first or the second exam were excluded (N = 4,679). In addition, we excluded patients who had any of the following conditions in the first or the second exam: alcohol intake ≥ 30 g/day in men or ≥ 20 g/day in women (N = 2,332), positive hepatitis B surface antigen or anti-hepatitis C virus antibodies (N = 1,107), liver cirrhosis (N = 458), or a history of cancer (N = 1,204). Finally, we then further excluded participants without any additional follow-up after baseline visit (N = 3,291), and with missing data on alcohol intake (N = 3,406). Since study participants could have more than one exclusion criteria, the final sample was 11,260 (**Fig 1**). The study was approved by the Institutional Review Board of the Samsung Medical Center. Informed

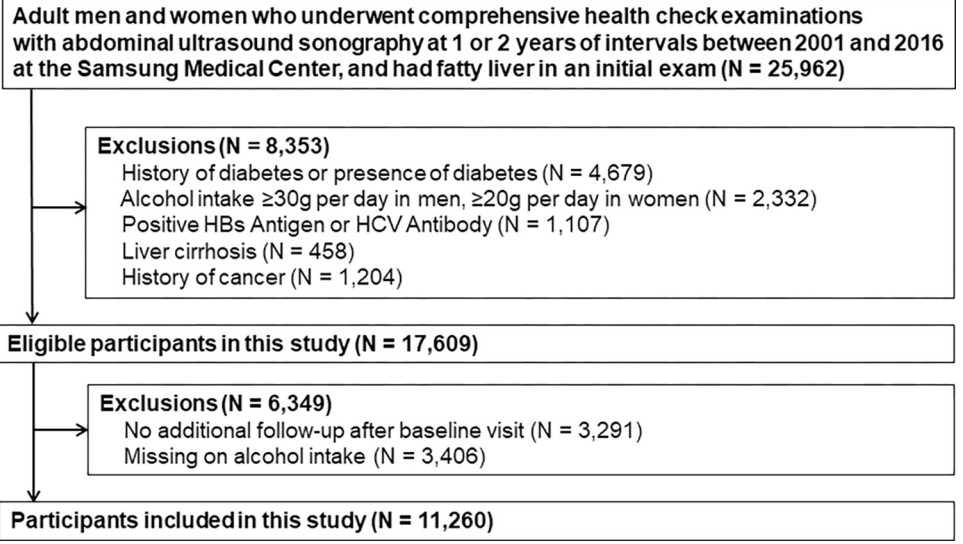

**Fig 1. Flowchart of study participants.**

consent was waived because the study was based on de-identified existing administrative and clinical data routinely collected for screening purposes.

## Data collection

**Abdominal US for NAFLD.** After optimizing technical parameters such as gain adjustment, placement of focal zone, and the optimum location of the transducer for each patient, abdominal US imaging was performed using LogiQ E9 (GE Healthcare, Milwaukee, WI, USA), iU22 xMatrix (Philips Medical Systems, Cleveland, OH, USA) or ACUSON Sequoia 512 devices (Siemens, Issaquah, WA, USA) by experienced radiologists unaware of the study aims [13]. The US was performed in a standard manner. The echogenicity of hepatic parenchyma was assessed and compared to the renal cortex at the mid-axillary line. Increased hepatorenal index, blurring of the portal vein wall, and marked attenuation of US beam that resulted in poor visualization of the diaphragm deep to the liver was considered hepatic steatosis [14,15]. Since we had already excluded participants with excessive alcohol use ($\geq$ 30 g/d in men and $\geq$ 20 g/d in women), as well as other identifiable causes of fatty liver between first and second exam as described in the exclusion criteria, fatty liver was considered NAFLD. NAFLD regression was defined as disappearance of fatty liver on abdominal US imaging at baseline (in the second exam).

Among participants with NAFLD, we calculated the NFS as -1.675 + 0.037 × age (years) + 0.094 × body mass index (BMI) (kg/m$^2$) + 1.13 × impaired fasting glucose/diabetes (yes = 1, no = 0) + 0.99 × AST/ALT ratio– 0.013 × platelet count (×10$^9$/l)– 0.66 × albumin (g/dl) [1]. NFS was used to assess the severity of fibrosis and to classify participants with NAFLD in two groups: high-intermediate (NFS $\geq$ -1.455) and low (NFS < -1.455) probability of advanced fibrosis [16].

**Diabetes.** Diabetes mellitus was defined as a fasting serum glucose $\geq$ 126 mg/dL, HbA1c $\geq$ 6.5%, a self-reported history of diabetes, or self-reported use of insulin or antidiabetic medications [17]. Blood specimens were sampled after at least a 12-hour fast. Glucose was measured as part of the health check-up exam at SMC's Department of Laboratory Medicine, which participates in several proficiency testing programs operated by the Korean

Association of Quality Assurance for Clinical Laboratory, the Asian Network of Clinical Laboratory Standardization and Harmonization, and the College of American Pathologists.

**Other variables.** Age at the health screening visit and sex were obtained from the electronic health record. Smoking status was categorized into never or ever smoker. Alcohol intake was categorized into none, light (< 20 g/day in men and < 10 g/day in women), and moderate (20 to < 30 g/day in men and 10 to < 20 g/day in women). Physical activity was categorized into < 3 times per week or ≥ 3 times per week.

Height, weight, waist circumference and sitting blood pressure were measured by trained nurses. BMI was calculated as weight in kilograms divided by height in meters squared and then classified according to Asian-specific criteria (underweight, BMI <18.5 kg/m$^2$; normal weight, BMI 18.5 to 22.9 kg/m$^2$; overweight, BMI 23 to 24.9 kg/m$^2$; and obese, BMI ≥25 kg/m$^2$) [18]. Blood pressure was measured using a mercury sphygmomanometer after the subject had been seated for at least 10 minutes. Hypertension was defined as systolic blood pressure ≥ 140 mmHg, diastolic blood pressure ≥ 90 mmHg, or the use of antihypertensive medication [17].

Serum levels of total cholesterol, low-density lipoprotein cholesterol (LDL-C), high-density lipoprotein cholesterol (HDL-C), and triglycerides (TG) were also measured as part of the health check-up exam at SMC's Department of Laboratory Medicine. Hyperlipidemia was defined as HDL-C < 40 mg/dl in men or < 50 mg/dl in women, TG ≥ 150 mg/dl, or the use of lipid-lowering medication [17].

## Statistical analyses

In this study, we conducted a descriptive analysis to compare the characteristics of study participants at the first and second exams by regressed and persistent NAFLD. To compare the characteristics, we calculated the standard mean difference (SMD).

The primary endpoint of this study was the incidence of diabetes. We followed the participants from the date of their second health screening visit (baseline) until the date of diagnosis of diabetes, the date of death, or the date of their last available follow-up visit, whichever came first. To analyze the incidence of diabetes, we calculated the cumulative incidence using Kaplan Meier methods. We calculated the incidence rate using the number of events divided by person-years. We used a proportional hazards regression model to estimate the hazard ratios (HRs) with 95% confidence intervals (CIs) for the development of diabetes. To control for potential confounding factors, we used a multivariable cox regression model. The adjusted model was controlled using age, sex, alcohol intake, smoking status, physical activity, BMI, hypertension, and hyperlipidemia at baseline. In the adjusted model, missing values in covariate were treated as a separate category by itself. We created another category for the missing values and use them as a different level.

In addition, we examined the association between regression of NAFLD and incident diabetes separately in pre-defined subgroups defined by age (< 50 and ≥ 50 years), sex (men and women), BMI (under-normal weight, overweight, and obese), smoking (never, and ever), alcohol drinking (none, light, and moderate), physical activity (< 3 times per week or ≥ 3 times per week), hypertension (no and yes), and hyperlipidemia (no and yes). Furthermore, we conducted a subgroup analysis based on the severity of fibrosis, dividing participants into two groups: those with low NFS (< -1.455) in the first exam and those with moderate and high NFS. Statistical analyses were performed with Stata version 16.0 (StataCorp LP, College Station, Texas) and R 3.2.1 (Vienna, Austria; http://www.R-project.org/). All reported p values are 2-tailed, and comparisons with p < 0.05 were considered statistically significant.

## Results

The mean (SD) age of study participants was 51.1 (7.8) years and 69.9% were men. Among the 11,260 NAFLD patients, 2,559 participants (22.7%) experienced regression of NAFLD (Table 1). Compared to participants with persistent NAFLD, those with regressed NAFLD were older, more likely to be women, never smokers, drinking none or light amount of alcohol, more physically active, and metabolically healthy both in the first and the second exams (baseline visit) (Table 1).

During 51,388 person-years of follow-up (median 4 years), 1,768 participants developed diabetes (341 and 1,427 in regressed and persistent group, respectively). The median length of time between the second abdominal US exam (baseline) and the diagnosis of diabetes among patients who incident diabetes was 3.6 years. The cumulative incidence of diabetes was consistently lower in participants with regressed NAFLD compared to those with persistent NAFLD over the entire follow-up (Fig 2).

After adjusting for age, sex, alcohol intake, smoking status, physical activity, BMI, hypertension, and hyperlipidemia at baseline, the HRs for incident diabetes in participants with regressed NAFLD compared to those with persistent NAFLD were 0.81 (95% CI 0.72–0.92) (Table 2).

When assessed by severity of fibrosis, among participants with low NFS ($< -1.455$) in the first exam, participants with regressed NAFLD had a lower risk of incident diabetes than those with persistent NAFLD (HR = 0.77, 95% CI = 0.68–0.88). However, in participants with intermediate to high NFS ($\geq -1.455$) in the first exam, the risk of incident diabetes was not different between NAFLD regression and persistence group (HR = 1.12, 95% CI = 0.82–1.51) (Table 3).

**Table 1. Characteristics of study participants by change of nonalcoholic fatty liver disease status ($N$ = 12,264).**

| Characteristics | In the first exam | | SMD | In the second exam (baseline) | | SMD |
|---|---|---|---|---|---|---|
| | Regressed (N = 2,559) | Persistent (N = 8,701) | | Regressed (N = 2,559) | Persistent (N = 8,701) | |
| Age (years) | 50.6 (7.9) | 49.8 (7.8) | -0.10 | 51.8 (7.9) | 51.0 (7.8) | -0.10 |
| Sex | | | -0.09 | | | -0.09 |
| Men | 1,613 (63.0) | 6,260 (72.0) | | 1,613 (63.0) | 6,260 (72.0) | |
| Women | 946 (37.0) | 2,441 (28.1) | | 946 (37.0) | 2,441 (28.1) | |
| BMI (kg/m$^2$) | 24.2 (3.0) | 25.3 (2.9) | 0.37 | 23.9 (2.5) | 25.4 (2.7) | 0.54 |
| Smoking | | | | | | |
| Never | 1,238 (48.4) | 3,601 (41.4) | -0.07 | 1,241 (48.5) | 3,661 (42.1) | -0.06 |
| Ever | 1,285 (50.2) | 4,978 (57.2) | 0.07 | 1,276 (49.9) | 4,922 (56.6) | 0.07 |
| Missing | 36 (1.4) | 122 (1.4) | -0.00 | 42 (1.6) | 118 (1.4) | -0.00 |
| Alcohol intake | | | | | | |
| None | 793 (31.0) | 2,336 (26.9) | -0.04 | 785 (30.7) | 2,293 (26.4) | -0.04 |
| Light | 1,444 (56.4) | 5,105 (58.7) | 0.02 | 1,468 (57.4) | 5,191 (59.7) | 0.02 |
| Moderate | 322 (12.6) | 1,260 (14.5) | 0.02 | 306 (12.0) | 1,217 (14.0) | 0.02 |
| Physical activity (per week) | | | | | | |
| Less than 3 times | 1,047 (40.9) | 4,022 (46.2) | 0.05 | 997 (39.0) | 3,995 (45.9) | 0.07 |
| 3 times or more | 1,234 (48.2) | 3,640 (41.8) | -0.06 | 1,377 (53.8) | 3,971 (45.6) | -0.08 |
| Missing | 278 (10.9) | 1,039 (11.9) | 0.01 | 185 (7.2) | 735 (8.5) | 0.01 |
| Hypertension | 661 (25.8) | 2,21 (32.4) | 0.07 | 672 (26.3) | 2,981 (34.3) | 0.08 |
| Hyperlipidemia | 1,227 (48.0) | 5,163 (59.3) | 0.11 | 1,192 (46.6) | 5,260 (60.5) | 0.14 |

Abbreviation: NAFLD, nonalcoholic fatty liver disease; BMI, body mass index; SMD, standard mean difference. Values are mean (SD), or number (%).

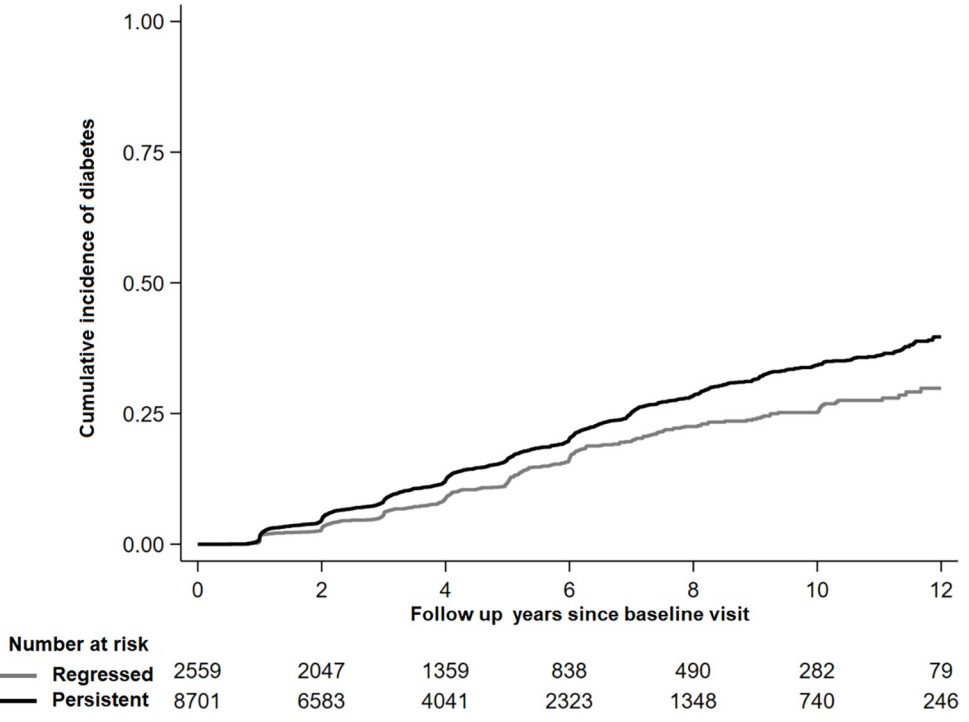

**Fig 2. Cumulative incidence of diabetes by non-alcoholic fatty liver disease regression status.**

In other pre-specified subgroups, the negative association between NAFLD regression and incident diabetes was consistent in all subgroups analyzed (all *p*-values for interaction > 0.10; **Fig 3**).

## Discussion

In this study, we demonstrated that regression of NAFLD was associated with reduced risk of incident diabetes in individuals with NAFLD. This finding indicates that regression of NAFLD may normalize metabolic abnormality associated with NAFLD, leading to decreased risk of incident diabetes. Notably, the association between regression of NAFLD and reduced risk of incident diabetes was evident for individuals with low probability of liver fibrosis, defined by low NFS (< -1.455), but not for individuals with intermediate to high probability of liver fibrosis, defined by intermediate to high NFS (≥ -1.455). This finding suggests that metabolic benefit of NAFLD regression might be attenuated when NAFLD patients has liver fibrosis. Thus, to maximize metabolic benefit from NAFLD regression, early intervention for NAFLD regression should be pursued before NAFLD patients has advanced fibrosis.

**Table 2. Hazard ratios (95% confidence intervals) for incident diabetes associated with nonalcoholic fatty liver disease regression (*N* = 11,260).**

| | Person- years | No. of cases | Incidence rate of diabetes (per 10000 person-years) | HR (95% CI) | |
| --- | --- | --- | --- | --- | --- |
| | | | | Crude | Adjusted |
| **NAFLD status** | | | | | |
| Regressed | 12,583 | 341 | 271 | 0.72 (0.64–0.81) | 0.81 (0.72–0.92) |
| Persistent | 38,805 | 1,427 | 368 | *Reference* | *Reference* |

Abbreviation: HR, hazard ratio; CI, confidence interval; NAFLD, nonalcoholic fatty liver disease.

Adjusted for age, sex, alcohol intake, smoking status, physical activity, body mass index, hypertension, and hyperlipidemia at baseline visit.

**Table 3. Hazard ratios (95% confidence intervals) for incident diabetes associated with nonalcoholic fatty liver disease regression by nonalcoholic fatty liver disease fibrosis score in the first exam (N = 11,254)*.**

| | Person-years | No. of cases | Incidence rate of diabetes (per 10000 person-years) | HR (95% CI) | |
| --- | --- | --- | --- | --- | --- |
| | | | | Crude | Adjusted |
| **NFS < -1.455 (N = 10,157)** | | | | | |
| Regressed | 11,217 | 283 | 252 | 0.69 (0.60–0.78) | 0.77 (0.68–0.88) |
| Persistent | 35,239 | 1,266 | 359 | *Reference* | *Reference* |
| **NFS ≥ -1.455 (N = 1,097)** | | | | | |
| Regressed | 1,349 | 58 | 430 | 0.95 (0.70–1.28) | 1.12 (0.82–1.51) |
| Persistent | 3,546 | 160 | 451 | *Reference* | *Reference* |
| **P for interaction** | | | | *0.05* | *0.03* |

Abbreviation: HR, hazard ratio; CI, confidence interval; NFS, nonalcoholic fatty liver disease fibrosis score.

Adjusted for age, sex, alcohol intake, smoking status, physical activity, body mass index, hypertension, and hyperlipidemia at baseline visit.

*6 patients were excluded due to missing on the NFS.

In terms of risk of incident diabetes, previous studies had conflicting findings. In a retrospective cohort study of 13,218 people without diabetes from a Korean occupational cohort, resolution of fatty liver by US, observed in 828 subjects, was not associated with a risk of incident diabetes [adjusted odds ratio 0.95 (95% CI 0.46,1.96)] [19]. In contrast, in a study of 2,726 subjects who had health check-up, NAFLD regression, observed in 155 subjects, showed similar risk of incident diabetes (HR 0.44, 95% CI 0.16, 1.20) compared to no NAFLD, while risk of incident diabetes was higher for subjects with incident NAFLD (HR 2.31, 95% CI 1.22, 4.36) or persistent NAFLD (HR 2.32, 95% CI 1.30, 4.12) [20]. In a study of 1940 men who had multiple health checkups, transient remission of NAFLD, observed in 139 men, showed lower HR of incident diabetes (HR 2.12, 95% CI 1.22, 3.57) than NAFLD persistent group (HR 3.44, 95% CI 2.29,5.21) when compared to subjects without NAFLD [21]. In the present study, we had a relatively large number of patients with NAFLD regression (n = 2,559) with a median 4 years of follow-up. We found that NAFLD regression was associated with decreased risk of incident diabetes.

The liver constitutes a key organ in systemic metabolism, contributing substantially to the development of insulin resistance [22]. NAFLD patients are commonly insulin-resistant [22,23]. Insulin resistance is a predictor of future development of type 2 diabetes [24]. Previously, only a few studies reported the metabolic effect of reversal of NAFLD [25,26]. In a study of 37 NASH (non-alcoholic steatohepatitis) patients who underwent metabolic surgery, remission of NASH was linked to reversal of insulin resistance [25]. Five years after the operation, NASH had reversed in 56.5% of the patients. The Homeostasis Model Assessment of Insulin Resistance (HOMA-IR) decreased from 3.31 ± 1.72 at baseline to 1.73 ± 1.08 (P < .001) after surgery. No significant differences in BMI or clinical parameters changes explained the effect of surgery on NASH, apart from the measure of insulin sensitivity post-surgery. A study of 164 NAFLD patients with impaired glucose tolerance showed that the disappearance of fatty liver after five years was associated with the normalization of glucose regulation [26]. Taken together with our observation, these findings suggest that the regression of NAFLD may improve insulin resistance and glucose dysregulation.

In subgroup analysis, the risk of incident diabetes decreased regardless of age, sex, BMI, smoking status, alcohol intake, physical activity, hypertension, and hyperlipidemia. However, the decreased risk of incident diabetes was observed only for NAFLD with low NFS (< -1.455), but not with intermediate to high NFS (≥ -1.455). NAFLD includes two pathologically distinct conditions with different prognoses: non-alcoholic fatty liver (NAFL) and

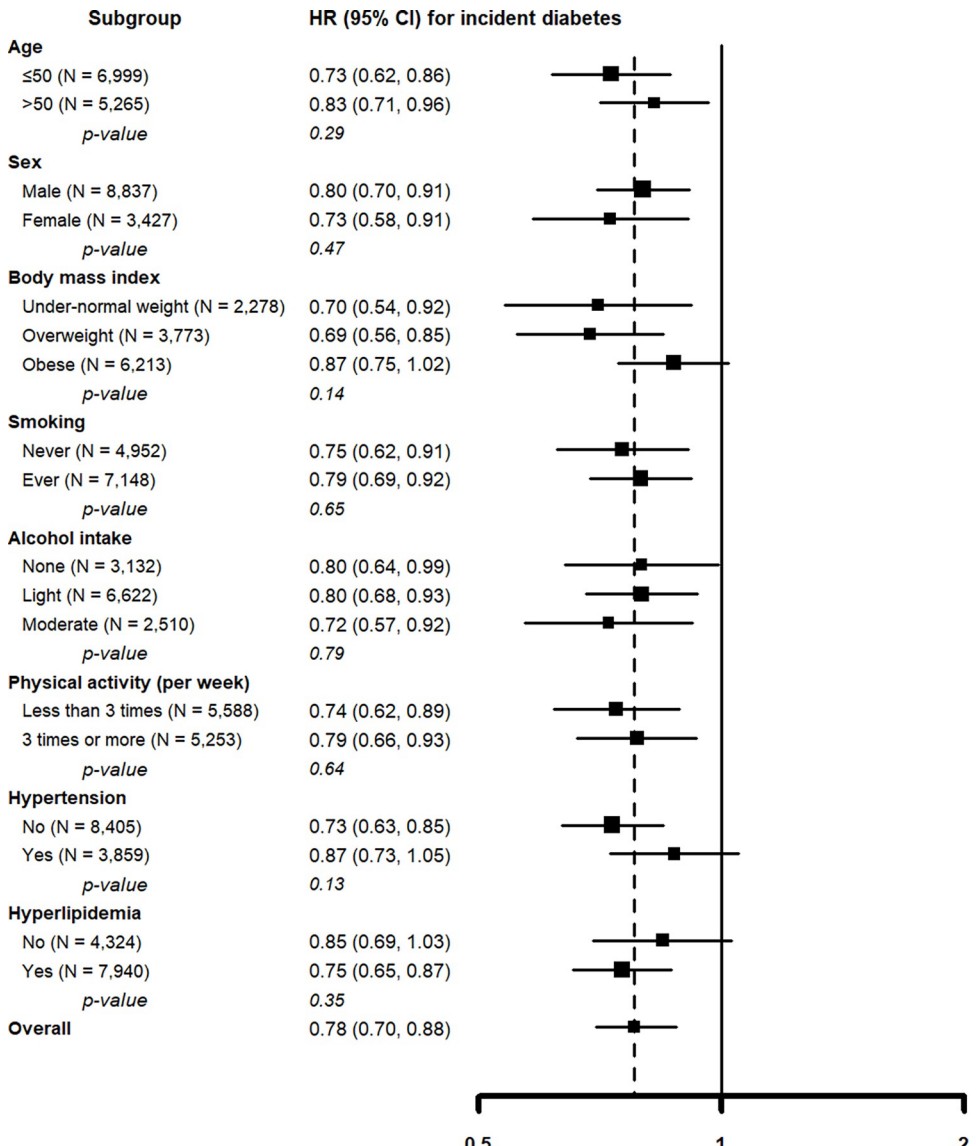

**Fig 3. Hazard ratios for incident diabetes comparing participants with regressed non-alcoholic fatty liver disease to those with persistent non-alcoholic fatty liver disease in predefined subgroups in the first exam.** Adjusted for age, sex, alcohol intake, smoking status, physical activity, body mass index, hypertension, and hyperlipidemia at baseline visit.

NASH. NASH differs from NAFL by necro-inflammatory response and fibrosis [27]. NASH patients have more severe adipose tissue insulin resistance and progressive reduction in whole-body insulin clearance compared to those with simple steatosis [28]. Although exact mechanism must be studied further, our finding suggest metabolic benefit by regression of NAFLD might be attenuated in NASH with fibrosis. Further studies are required to see whether metabolic benefit is limited for individuals with NAFL or not.

This study has some limitations. Fatty liver was defined by US, which might have measurement errors [29]. During the study period, many radiologists were involved in performing abdominal US, which might have led to inter- and intra-observer variability. In addition, US is a sensitive but imperfect tool to diagnose hepatic steatosis [30]. The gold standard for

diagnosing hepatic steatosis is liver biopsy. Hence, our data needs validation using histology to diagnose the presence of hepatic steatosis. We used landmark period (1 or 2 years from the first US exam) to assess the change of NAFLD status. However, NAFLD status might also have changed after baseline visit. This study was performed in Asian population and might not be applied to other ethnicities. The strength of this study includes the longitudinal design and the large sample size.

In conclusion, we demonstrated that regression of NAFLD was associated with decreased risk of incident diabetes compared to persistent NAFLD. This finding suggests that interventions focused on promoting NAFLD regression would be an effective strategy to decrease the burden of diabetes. The benefit was evident when NAFLD patients had low NFS, which suggests that early intervention for NAFLD is required before progression to NASH, to maximize metabolic benefit from NAFLD regression. In clinical practice, NAFLD with low NFS may be overlooked due to the perception of the condition as relatively benign. However, this may be the optimal time for NAFLD intervention, as NAFLD regression at this point can have a more significant impact on reducing the risk of diabetes. Prospective interventional trials are warranted to investigate whether and when NAFLD patients can benefit from NAFLD management, e.g. lifestyle modification or pharmacological treatment, in preventing diabetes or other metabolic complications.

## Author Contributions

**Conceptualization:** Dong Hyun Sinn, Geum-Youn Gwak.

**Data curation:** Danbee Kang, Sung Chul Choi.

**Formal analysis:** Danbee Kang.

**Investigation:** Dong Hyun Sinn, Eliseo Guallar, Juhee Cho.

**Methodology:** Yun Soo Hong, Juhee Cho.

**Project administration:** Geum-Youn Gwak.

**Resources:** Sung Chul Choi.

**Software:** Danbee Kang.

**Supervision:** Eliseo Guallar, Juhee Cho, Geum-Youn Gwak.

**Validation:** Danbee Kang, Eliseo Guallar, Juhee Cho.

**Visualization:** Danbee Kang.

**Writing – original draft:** Dong Hyun Sinn, Danbee Kang.

**Writing – review & editing:** Eliseo Guallar, Yun Soo Hong, Yewan Park, Juhee Cho, Geum-Youn Gwak.

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
