## [Decision Letter · Decision Letter 0]

2 May 2023

PONE-D-23-02839Regression of nonalcoholic fatty liver disease is associated with reduced risk of incident diabetes: A longitudinal cohort studyPLOS ONE

Dear Dr. Gwak,

Thank you for submitting your manuscript to PLOS ONE. After careful consideration, we feel that it has merit but does not fully meet PLOS ONE’s publication criteria as it currently stands. Therefore, we invite you to submit a revised version of the manuscript that addresses the points raised during the review process.

We look forward to receiving your revised manuscript.

Kind regards,

Ahmed Mustafa Rashid

Academic Editor

PLOS ONE

- https://www.karger.com/Article/FullText/443344?

- https://academic.oup.com/jcem/article/106/3/750/6009066?login=false

- https://eje.bioscientifica.com/configurable/content/journals$002feje$002f181$002f2$002fEJE-19-0143.xml? :ac=journals%24002feje%24002f181%24002f2%24002fEJE-19-0143.xml

In your revision ensure you cite all your sources (including your own works), and quote or rephrase any duplicated text outside the methods section. Further consideration is dependent on these concerns being addressed.

Reviewers' comments:

Reviewer's Responses to Questions

**Comments to the Author**

1. Is the manuscript technically sound, and do the data support the conclusions?

Reviewer #1: Yes

Reviewer #2: Yes

2. Has the statistical analysis been performed appropriately and rigorously? 

Reviewer #1: Yes

Reviewer #2: Yes

3. Have the authors made all data underlying the findings in their manuscript fully available?

Reviewer #1: No

Reviewer #2: Yes

4. Is the manuscript presented in an intelligible fashion and written in standard English?

Reviewer #1: Yes

Reviewer #2: Yes

5. Review Comments to the Author

Reviewer #1: The data in the manuscript support the conclusion, and the methodology appears to be correct. Although the statistical analysis has been performed accurately, there should be a more detailed discussion of the statistical methods used. Furthermore, the authors have not fully made all the data underlying their findings available, with some restrictions mentioned. The manuscript is in standard English language, but it will benefit from a few grammatical corrections.

Reviewer #2: Sinn D.H et al conducted a retrospective longitudinal cohort study titled “Regression of nonalcoholic fatty liver disease is associated with reduced risk of incident diabetes: A longitudinal cohort study.” In this study the authors evaluated whether or not regression of NAFLD is associated with a reduction in development of diabetes. In my opinion the study can be improved by incorporating the following changes:

1. In the title of the manuscript all important terminologies such as nonalcoholic fatty liver disease and diabetes should start with capital letters (Nonalcoholic Fatty Liver Disease, Diabetes etc.)

2. Methods section, data collection portion, lines 141-143: while the authors have mentioned that the cut off value for NFS was -1.455, it would add value to the quality of the manuscript if the authors could cite a relevant reference for this. Similarly, relevant references for the definitions of diabetes, hypertension and hyperlipidemia should also be cited.

3. Discussion section, line 259: the full form of NASH should be mentioned as this is the first time this term has been used in the manuscript.

6. PLOS authors have the option to publish the peer review history of their article (what does this mean?). If published, this will include your full peer review and any attached files.

Reviewer #1: No

Reviewer #2: No

---

## [Author Response · Author response to Decision Letter 0]

7 Jun 2023

Response to the editor

General comments

Sinn et al. conducted a study on “Regression of nonalcoholic fatty liver disease is associated with reduced risk of incident diabetes: A longitudinal cohort study”, in which they investigated the association between regression of nonalcoholic fatty liver disease (NAFLD) and the risk of incident diabetes using a large cohort. The retrospective cohort study included 11,260 participants who underwent at least two health check examinations with abdominal ultrasound between 2001 and 2016 at the Samsung Medical Center. NAFLD regression was defined as disappearance of fatty liver on abdominal US imaging at baseline. Diabetes was defined as a fasting serum glucose ≥ 126 mg/dL, HbA1c ≥ 6.5%, a self-reported history of diabetes, or self-reported use of insulin or antidiabetic medications. The study found that NAFLD regression was associated with a lower risk of incident diabetes.

Specific comment: 

From my perspective, the inclusion of the following revisions could enhance this study:

Thank you

Abstract: 

1. Include the duration of the study in the abstract, which is important for readers to understand the scope of the study.

As the reviewer recommended, we now included the duration of the study in the Abstract (page 4, lines 72-74).

“A cohort of 11,260 adults who had NAFLD at an initial exam and had the second evaluation for NAFLD status at 1~2 years from the initial exam were followed up for incident diabetes from 2001 to 2016.”

2. In the first sentence of the abstract, "whether improvement of NAFLD leads to clinical benefit remains uncertain" should be corrected to "whether improvement of NAFLD leads to clinical benefits remains uncertain." (Page 4, Line 60)

Done

3. Also in methods section, "A cohort of 11,260 adults who had NAFLD in an initial exam" should be corrected to "A cohort of 11,260 adults who had NAFLD at an initial exam." (Page 4, Line 64)

Done

4. In the results section, "The fully adjusted hazard ratio (HR) for incident diabetes in participants with regressed NAFLD compared to those with persistent NAFLD were 0.81" should be corrected to "The fully adjusted hazard ratio (HR) for incident diabetes in participants with regressed NAFLD compared to those with persistent NAFLD was 0.81." (Page 4, Line 72)

Done

5. In the results section, "When assessed by NAFLD severity, among participants with low NAFLD fibrosis score (NFS) (< -1.455), participants with regressed NAFLD had a lower risk of incident diabetes than those with persistent NAFLD" should be corrected to "When assessed by NAFLD severity, among participants with a low NAFLD fibrosis score (NFS) (< -1.455), participants with regressed NAFLD had a lower risk of incident diabetes than those with persistent NAFLD." (Page 4, Line 73)

Done

6. In the results section, "However, in participants with intermediate to high NFS (≥ -1.455), the risk of incident diabetes was not different between NAFLD regression and persistence group" should be corrected to "However, in participants with an intermediate to high NFS (≥ -1.455), the risk of incident diabetes was not different between the NAFLD regression and persistence groups." (Page 4, Line 76,77)

Done

7. In the conclusions section, "This suggests that early intervention for NAFLD, before advanced fibrosis is present, may maximize metabolic benefit from NAFLD regression" should be corrected to "This suggests that early intervention for NAFLD, before advanced fibrosis is present, may maximize the metabolic benefit from NAFLD regression." (Page 5, Line 82)

Done

Introduction: 

8. The first sentence in the introduction should be revised to make it clearer. For example, "Nonalcoholic fatty liver disease (NAFLD) is a condition in which the liver accumulates fat without significant alcohol intake, viral hepatitis, medications that would cause fatty liver, or other obvious causes." (Page 6, Line 84)

With respect to the editor’s comment, we now revised the sentence as follow (page 5, line 92-94)

“Nonalcoholic fatty liver disease (NAFLD) is a condition in which the liver accumulates fat without significant alcohol intake, viral hepatitis, medications that would cause fatty liver, or other obvious causes.”

9. In the introduction section, the authors should provide more information on the prevalence of NAFLD and its impact on public health.

As the editor requested, we included more information on the prevalence of NAFLD and its impact on public health (page 5, line 95)

“NAFLD is the most common chronic liver disease, with a worldwide prevalence of 25% [2]. NAFLD patients are at an increased risk of adverse outcomes, including overall mortality, and liver-specific morbidity and mortality, [3] and are projected to continue to increase, which has an important impact on public health [4].”

10. The research question is not clear, and it is not stated explicitly in the introduction section. The authors should state the research question clearly, for example: “Is regression of NAFLD associated with a decreased risk of incident diabetes?”

In response to the editor’s comment, we revised the sentence in the Introduction to make the research question clearer (page 5, lines 105-106)

“However, to date, whether regression of NAFLD is associated with a decreased risk of incident diabetes is largely unexplored.” 

Methods: 

11. The number of excluded participants in both the text and Figure 1 does not add up to 11,260. (Page 7, Line 117)

Since study participants could have more than one exclusion criterion, the final sample was 11,260. We have revised the sentence to clarify this point.

Methods (page 6, lines 126-127) 

“Since study participants could have more than one exclusion criteria, the final sample was 11,260 (Fig 1).”

12. The statistical methods used for the analysis should be described in more detail. The authors should provide more information on how they conducted the analysis, including the statistical tests used and how they controlled for potential confounding factors.

In response to the editor’s comments, we added more information on statistical analysis parts in the Methods section as follows: 

Methods (page 8-9, lines 187-212) 

“In this study, we conducted a descriptive analysis to compare the characteristics of study participants at the first and second exams by regressed and persistent NAFLD. To compare the characteristics, we calculated the standard mean difference (SMD).

The primary endpoint of this study was the incidence of diabetes. We followed the participants from the date of their second health screening visit (baseline) until the date of diagnosis of diabetes, the date of death, or the date of their last available follow-up visit, whichever came first. To analyze the incidence of diabetes, we calculated the cumulative incidence using Kaplan Meier methods. We calculated the incidence rate using the number of events divided by person-years. We used a proportional hazards regression model to estimate the hazard ratios (HRs) with 95% confidence intervals (CIs) for the development of diabetes. To control for potential confounding factors, we used a multivariable cox regression model. The adjusted model was controlled using age, sex, alcohol intake, smoking status, physical activity, BMI, hypertension, and hyperlipidemia at baseline. In the adjusted model, missing values in the covariate were treated as a separate category by itself. We created another category for the missing values and use them as a different level. 

In addition, we examined the association between regression of NAFLD and incident diabetes separately in pre-defined subgroups defined by age (< 50 and ≥ 50 years), sex (men and women), BMI (under-normal weight, overweight, and obese), smoking (never, and ever), alcohol drinking (none, light, and moderate), physical activity (< 3 times per week or ≥ 3 times per week), hypertension (no and yes), and hyperlipidemia (no and yes). Furthermore, we conducted a subgroup analysis based on the severity of fibrosis, dividing participants into two groups: those with low NFS (< -1.455) in the first exam and those with moderate and high NFS. 

Statistical analyses were performed with Stata version 16.0 (StataCorp LP, College Station, Texas) and R 3.2.1 (Vienna, Austria; http://www.R-project.org/). All reported p values are 2-tailed, and comparisons with P < 0.05 were considered statistically significant.”

13. In the methods section, the authors should describe the characteristics of the study population in more detail, including their age, sex, and other relevant demographic information.

Age at the health screening visit and sex were obtained from the electronic medical record, while other relevant information was collected through self-reported questionnaires or examinations. The detailed characteristics of the study population, including age, sex, and other relevant demographic information, are described in the Results section (page 8, line 167) and Table 1.

14. In the methods section, the authors should provide more information on the follow-up period and the time between the second abdominal US exam and the diagnosis of diabetes.

The primary endpoint of this study was the incidence of diabetes. We followed the participants from the date of their second health screening visit (baseline) until the date of diagnosis of diabetes, the date of death, or the date of their last available follow-up visit, whichever came first. During 51,388 person-years of follow-up (median 4 years), 1,768 participants developed diabetes. Among the participants who developed diabetes, the median time interval between the second abdominal US exam (baseline) and the diagnosis of diabetes was 3.6 years. With respect to the editor’s comment, we included more information on the follow-up period and the length of time between the second abdominal US exam and the diagnosis of diabetes (page 9, lines 191-193). 

“We followed the participants from the date of their second health screening visit (baseline) until the date of their initial diagnosis of diabetes, the date of death, or the date of their last available follow-up visit, whichever came first.”

In the Results section, we have added the following sentence (page 11, lines 227-230): 

“During 51,388 person-years of follow-up (median 4 years), 1,768 participants developed diabetes. The median length of time between the second abdominal US exam (baseline) and the diagnosis of diabetes among participants who had incident diabetes was 3.6 years.”

15. In the methods section, the authors should describe how missing data were handled, particularly for variables such as alcohol intake.

In the adjusted model, missing values in covariate were treated as a separate category by itself. We created another category for the missing values and use them as a different level. However, alcohol intake is important to define NAFLD, thus, we excluded 3,406 participants who had missing on the alcohol intake variable on either the first or second exam. We had described how we handled missing variable in the Methods section (page 6, lines 125-126, page 9, lines 199-201). 

“Finally, we then further excluded participants without any additional follow-up after baseline visit (N =3,291), and with missing data on alcohol intake (N = 3,406).”

“In the adjusted model, missing values in covariate were treated as a separate category by itself. We created another category for the missing values and use them as a different level.”

16. The description of how the abdominal ultrasound imaging was performed could be clearer. The type of equipment used for imaging is listed, but the imaging protocols and criteria used to diagnose NAFLD should also be explained.

With respect to the editor’s comment, we revised the Methods section to make this point clearer (page 8, lines 136-149)

“After optimizing technical parameters such as gain adjustment, placement of focal zone, and the optimum location of the transducer for each participant, abdominal US imaging was performed using LogiQ E9 (GE Healthcare, Milwaukee, WI, USA), iU22 xMatrix (Philips Medical Systems, Cleveland, OH, USA) or ACUSON Sequoia 512 devices (Siemens, Issaquah, WA, USA) by experienced radiologists unaware of the study aims [13]. The US was performed in a standard manner. The echogenicity of hepatic parenchyma was assessed and compared to the renal cortex at the mid-axillary line. Increased hepatorenal index, blurring of the portal vein wall, and marked attenuation of US beam that resulted in poor visualization of the diaphragm deep to the liver was considered hepatic steatosis [14,15].

Results: 

17. In the results section, the authors should present the results in a more organized and clear manner. It is not clear how many participants had regression of NAFLD, and how many developed incident diabetes.

Among the 11,260 NAFLD patients, 2,559 (22.7%) experienced regression of NAFLD between the first and second (baseline) health screening visit. When we followed them up (median 4 years), 1,768 participants developed diabetes (341 and 1,427 in the regressed and persistent group, respectively). In response to the editor’s comment, we revised the sentences in the Results sections as follows. 

Result (page 10, lines 215-216)

“Among the 11,260 NAFLD patients, 2,559 participants (22.7%) experienced regression of NAFLD (Table 1).” 

Result (page 11, lines 227-228)

“During 51,388 person-years of follow-up (median 4 years), 1,768 participants developed diabetes (341 and 1,427 in regressed and persistent group, respectively).” 

Discussion: 

18. The authors could provide more context for their findings by discussing how their results relate to previous studies on the association between NAFLD regression and incident diabetes. Additionally, the authors should discuss the potential limitations of the study, such as the use of abdominal ultrasound to diagnose NAFLD, which may not be as accurate as liver biopsy.

With respect to this comment, we have revised the Discussion section as follows. 

Discussion (page 14, lines 284-297)

“In terms of risk of incident diabetes, previous studies had conflicting findings. In a retrospective cohort study of 13,218 people without diabetes from a Korean occupational cohort, resolution of fatty liver by US, observed in 828 subjects, was not associated with a risk of incident diabetes [adjusted odds ratio 0.95 (95% CI 0.46,1.96)]. In contrast, in a study of 2,726 subjects who had health check-up, NAFLD regression, observed in 155 subjects, showed similar risk of incident diabetes (HR 0.44, 95% CI 0.16, 1.20) compared to no NAFLD, while risk of incident diabetes was higher for subjects with incident NAFLD (HR 2.31, 95% CI 1.22, 4.36) or persistent NAFLD (HR 2.32, 95% CI 1.30, 4.12) [18]. In a study of 1,940 men who had multiple health checkups, transient remission of NAFLD, observed in 139 men, showed a lower HR of incident diabetes (HR 2.12, 95% CI 1.22, 3.57) than NAFLD persistent group (HR 3.44, 95% CI 2.29,5.21) when compared to subjects without NAFLD [19]. In the present study, we had a relatively large number of patients with NAFLD regression (n = 2,559) with a median 4 years of follow-up. We found that NAFLD regression was associated with decreased risk of incident diabetes.”

Discussion (page 15, lines 325-328) 

“In addition, US is a sensitive but imperfect tool to diagnose hepatic steatosis [28]. The gold standard for diagnosing hepatic steatosis is liver biopsy. Hence, our data needs validation using histology to diagnose the presence of hepatic steatosis.”

Conclusion:

19. In the discussion section, the authors should provide more information on the implications of the study findings and how they relate to previous research in this area.

In response to the editor’s comment, we revised the Conclusion section as follows (page 16, lines 333-344)

“In conclusion, we demonstrated that regression of NAFLD was associated with decreased risk of incident diabetes compared to persistent NAFLD. This finding suggests that interventions focused on promoting NAFLD regression would be an effective strategy to decrease the burden of diabetes. The benefit was evident when NAFLD patients had low NFS, which suggests that early intervention for NAFLD is required before progression to NASH, to maximize metabolic benefit from NAFLD regression. In clinical practice, NAFLD with low NFS may be overlooked due to the perception of the condition as relatively benign. However, this may be the optimal time for NAFLD intervention, as NAFLD regression at this point can have a more significant impact on reducing the risk of diabetes. Prospective interventional trials are warranted to investigate whether and when NAFLD patients can benefit from NAFLD management, e.g. lifestyle modification or pharmacological treatment, in preventing diabetes or other metabolic complications.”

 

Response to the Reviewer 1

1. The data in the manuscript support the conclusion, and the methodology appears to be correct. Although the statistical analysis has been performed accurately, there should be a more detailed discussion of the statistical methods used. Furthermore, the authors have not fully made all the data underlying their findings available, with some restrictions mentioned. The manuscript is in standard English language, but it will benefit from a few grammatical corrections.

In response to the reviewer’s comments, we revised the statistical analysis parts in the Methods section as follows: 

Methods (page 8, lines 187-212) 

“In this study, we conducted a descriptive analysis to compare the characteristics of study participants at the first and second exams by regressed and persistent NAFLD. To compare the characteristics, we calculated the standard mean difference (SMD).

The primary endpoint of this study was the incidence of diabetes. We followed the participants from the date of their second health screening visit (baseline) until the date of diagnosis of diabetes, the date of death, or the date of their last available follow-up visit, whichever came first. To analyze the incidence of diabetes, we calculated the cumulative incidence using Kaplan Meier methods. We calculated the incidence rate using the number of events divided by person-years. We used a proportional hazards regression model to estimate the hazard ratios (HRs) with 95% confidence intervals (CIs) for the development of diabetes. To control for potential confounding factors, we used a multivariable cox regression model. The adjusted model was controlled using age, sex, alcohol intake, smoking status, physical activity, BMI, hypertension, and hyperlipidemia at baseline. In the adjusted model, missing values in the covariate were treated as a separate category by itself. We created another category for the missing values and use them as a different level. 

In addition, we examined the association between regression of NAFLD and incident diabetes separately in pre-defined subgroups defined by age (< 50 and ≥ 50 years), sex (men and women), BMI (under-normal weight, overweight, and obese), smoking (never, and ever), alcohol drinking (none, light, and moderate), physical activity (< 3 times per week or ≥ 3 times per week), hypertension (no and yes), and hyperlipidemia (no and yes). Furthermore, we conducted a subgroup analysis based on the severity of fibrosis, dividing participants into two groups: those with low NFS (< -1.455) in the first exam and those with moderate and high NFS. 

Statistical analyses were performed with Stata version 16.0 (StataCorp LP, College Station, Texas) and R 3.2.1 (Vienna, Austria; http://www.R-project.org/). All reported p values are 2-tailed, and comparisons with P < 0.05 were considered statistically significant.”

Response to the Reviewer 2

General comments: 

Sinn D.H et al conducted a retrospective longitudinal cohort study titled “Regression of nonalcoholic fatty liver disease is associated with reduced risk of incident diabetes: A longitudinal cohort study.” In this study the authors evaluated whether or not regression of NAFLD is associated with a reduction in development of diabetes. In my opinion the study can be improved by incorporating the following changes:

Thank you.

Specific comments: 

1. In the title of the manuscript all important terminologies such as nonalcoholic fatty liver disease and diabetes should start with capital letters (Nonalcoholic Fatty Liver Disease, Diabetes etc.)

Done. 

2. Methods section, data collection portion, lines 141-143: while the authors have mentioned that the cut off value for NFS was -1.455, it would add value to the quality of the manuscript if the authors could cite a relevant reference for this. Similarly, relevant references for the definitions of diabetes, hypertension and hyperlipidemia should also be cited.

As the reviewer suggested, we included the references of cut off value for NFS, and cited the definitions of diabetes, hypertension and hyperlipidemia in the Methods section. 

“Among participants with NAFLD, we calculated the NFS as -1.675 + 0.037 × age (years) + 0.094 × body mass index (BMI) (kg/m2) + 1.13 × impaired fasting glucose/diabetes (yes=1, no=0) + 0.99 × AST/ALT ratio – 0.013 × platelet count (×109/l) – 0.66 × albumin (g/dl) [1]. NFS was used to assess the severity of fibrosis and to classify participants with NAFLD in two groups: high-intermediate (NFS ≥ -1.455) and low (NFS < -1.455) probability of advanced fibrosis [16]. 

…

Diabetes mellitus was defined as a fasting serum glucose ≥ 126 mg/dL, HbA1c ≥ 6.5%, a self-reported history of diabetes, or self-reported use of insulin or antidiabetic medications [17]. 

…

Blood pressure was measured using a mercury sphygmomanometer after the subject had been seated for at least 10 minutes. Hypertension was defined as systolic blood pressure ≥ 140 mmHg, diastolic blood pressure ≥ 90 mmHg, or the use of antihypertensive medication [17].

Serum levels of total cholesterol, low-density lipoprotein cholesterol (LDL-C), high-density lipoprotein cholesterol (HDL-C), and triglycerides (TG) were also measured as part of the health check-up exam at SMC’s Department of Laboratory Medicine. Hyperlipidemia was defined as HDL-C < 40 mg/dl in men or < 50 mg/dl in women, TG ≥ 150 mg/dl, or the use of lipid-lowering medication [17].

3. Discussion section, line 259: the full form of NASH should be mentioned as this is the first time this term has been used in the manuscript.

Done

 

1. Please ensure that your manuscript meets PLOS ONE's style requirements, including those for file naming. The PLOS ONE style templates can be found at https://journals.plos.org/plosone/s/file?id=wjVg/PLOSOne_formatting_sample_main_body.pdf and https://journals.plos.org/plosone/s/file?id=ba62/PLOSOne_formatting_sample_title_authors_affiliations.pdf.

Done

- https://www.karger.com/Article/FullText/443344?

- https://academic.oup.com/jcem/article/106/3/750/6009066?login=false

- https://eje.bioscientifica.com/configurable/content/journals$002feje$002f181$002f2$002fEJE-19-0143.xml? :ac=journals%24002feje%24002f181%24002f2%24002fEJE-19-0143.xml

In your revision ensure you cite all your sources (including your own works), and quote or rephrase any duplicated text outside the methods section. Further consideration is dependent on these concerns being addressed.

We have cited the first two articles for this manuscript. The third one is an article that we have published using the same cohort, but with different study questions. As we used the same cohort data from the Samsung Medical Center Health Promotion Center for this study, details of data collection and abdominal US overlap with our previous study (PMID 31176297). We have cited relevant publications and revised the Method section to avoid duplicated text. 

The original submission already included the information that the informed consent was waived by the IRB in our institution. 

The data is available upon request. Please refer to the answer to Comment No 5.

The data contain potentially sensitive information that can be used to identify or specify individual. Also, the data policy in our institution regulates sharing a de-identified data set, as it can be used to identify or specify individual. Data are available upon request to Samsung Medical Center Institutional Data Access / Ethics Committee (dm.cha@samsung.com), after approval. Anyone who want to contact Samsung Medical Center Institutional Data/Ethics Committee, they can e-mail directly or e-mail to corresponding author for researchers who meet the criteria for access to confidential data.

The study was approved by the Institutional Review Board of the Samsung Medical Center. Informed consent was waived because the study was based on de-identified existing administrative and clinical data routinely collected for screening purposes. This information is described in the Method section.

---

## [Decision Letter · Decision Letter 1]

5 Jul 2023

Regression of Nonalcoholic Fatty Liver Disease is associated with reduced risk of incident Diabetes: A longitudinal cohort study

PONE-D-23-02839R1

Dear Dr. Gwak,

We’re pleased to inform you that your manuscript has been judged scientifically suitable for publication and will be formally accepted for publication once it meets all outstanding technical requirements.

Kind regards,

Ahmed Mustafa Rashid

Academic Editor

PLOS ONE

Additional Editor Comments (optional):

Reviewers' comments:

Reviewer's Responses to Questions

**Comments to the Author**

1. If the authors have adequately addressed your comments raised in a previous round of review and you feel that this manuscript is now acceptable for publication, you may indicate that here to bypass the “Comments to the Author” section, enter your conflict of interest statement in the “Confidential to Editor” section, and submit your "Accept" recommendation.

Reviewer #1: All comments have been addressed

Reviewer #2: (No Response)

2. Is the manuscript technically sound, and do the data support the conclusions?

Reviewer #1: Yes

Reviewer #2: Yes

3. Has the statistical analysis been performed appropriately and rigorously? 

Reviewer #1: Yes

Reviewer #2: Yes

4. Have the authors made all data underlying the findings in their manuscript fully available?

Reviewer #1: No

Reviewer #2: Yes

5. Is the manuscript presented in an intelligible fashion and written in standard English?

Reviewer #1: Yes

Reviewer #2: Yes

6. Review Comments to the Author

Reviewer #1: The authors have adequately addressed all the comments raised in the previous round of review, and the manuscript is now deemed acceptable for publication. The manuscript is technically sound, and the data provided support the conclusions drawn. The statistical analysis has been performed appropriately and rigorously. However, it should be noted that the authors have indicated that not all data underlying the findings in their manuscript are fully available. Furthermore, I have no concerns regarding dual publication, research ethics, or publication ethics.

Reviewer #2: (No Response)

7. PLOS authors have the option to publish the peer review history of their article (what does this mean?). If published, this will include your full peer review and any attached files.

Reviewer #1: No

Reviewer #2: No

---

## [Editor Report · Acceptance letter]

10 Jul 2023

PONE-D-23-02839R1 

Regression of Nonalcoholic Fatty Liver Disease is associated with reduced risk of incident Diabetes: A longitudinal cohort study 

Dear Dr. Gwak:

I'm pleased to inform you that your manuscript has been deemed suitable for publication in PLOS ONE. Congratulations! Your manuscript is now with our production department. 

Kind regards, 

on behalf of

Dr. Ahmed Mustafa Rashid 

Academic Editor

PLOS ONE